# ECG Signal Imputation using Reservoir Computing Energy Based Models

Anonymous Full Paper
Submission ###

## Abstract

Many biological time-series datasets (e.g. neural recordings and physiological signals) have missing segments due to sensor dropouts or noise, which leads to a need for a robust imputation method that also quantifies confidence. We propose an architecture using reservoir computing (Echo State Networks and Liquid State Machines) in series with an Energy-Based Model (EBM) to perform the imputation. An EBM models the underlying data distribution by assigning a scalar energy to possible outputs, with a low energy for plausible imputations and high energy for implausible ones. Therefore, the energy serves as a heuristic for uncertainty of the prediction, meaning unreliable imputations may be detected by their higher energy. To capture the complex temporal dynamics of biological data, we embed inputs into a high-dimensional state space using reservoir computing, either using Echo State Networks (ESNs) or Liquid State Machines (LSMs). ESNs have fixed random recurrent weights, and LSMs have spiking neurons. Both these reservoirs expand and filter the input sequences into rich nonlinear features, simplifying the imputation task for the EBM. The output of the reservoir is passed into the EBM using two different methods. Either raw signals from the reservoir are used as inputs or we perform a dimensionality reduction of the reservoir states, via principal component analysis (PCA), to identify low-dimensional manifolds of temporal patterns, which can denoise the reservoir representation to create a cleaner signal for the EBM. Preliminary results on ECG signals show that using an LSM as the reservoir or the lower dimensional manifold of an ESN produces the best features for the EBM to do signal imputation.

## 1 Introduction

Time-series data often contain missing values due to failures in data capture or noise. Imputing missing observations is thus a necessary task needed for the analysis of time series data downstream. However, it is important that we understand how reliable the reconstruction is given the absence of a ground truth. As such, models which quantify uncertainty in their outputs are advantageous. For example, in the case of physiological timeseries, a predicted segment of an electrocardiogram (ECG) time-series that the model finds unreliable should be treated with caution in analysis or omitted, which becomes especially important in clinical settings. Currently it is a serious limitation that conventional imputation methods lack the ability to quantify uncertainty.

One class of models which are able to report their uncertainty are Energy-Based Models (EBMs), which learn a scalar energy function over inputs and outputs [1-4]. EBMs are given an observed context x and a completion y, the energy $E(x,y)$ quantifies the compatibility of the input and output, and the inference selects low-energy completions. Incompatible pairings will be given a high energy, and more compatible pairs will have a lower energy score. Therefore, we can utilise the energy score as a risk indicator, as shown in [1–4]. If the lowest energy for a completion still remains high, this means that even the best imputation is implausible and therefore has a high uncertainty, which is particularly important for biological time series data, where high uncertainty should not be accepted. When considering how time series data is collected, there are numerous examples collected from edge sensors at high frequencies, making their processing computationally expensive. Reservoir computing (RC) has presented itself as a framework which has shown the ability to model complex time series with short training times on low power edge devices. To do this, RC utilises a fixed, large recurrent network to produce a high-dimensional state trajectory. The reservoir therefore suppresses noise from the input signal and acts as a filter that makes downstream prediction more linear and tractable [5,6]. A common form of reservoir computing model are echo state networks, which have a continuous activation (e.g., tanh) and produce smooth trajectories in their state space with fading memory [7-9]. Another commonly used reservoir model is the liquid state machine, which uses spiking neurons (e.g., leaky integrate-and-fire), which yield space event-driven "liquid" dynamics that are energy-efficient and provide neurophysiologically plausible learning [10-12]. Both cases of reservoirs transform the low-dimensional signal into a rich state sequence, making data linearly separable allowing linear readout models to be trained quickly over this state sequence.

Our innovation is therefore to combine EBMs with reservoir computing for efficient imputation of

a time series, while providing an uncertainty over this prediction. Functionally, EBMs deduce energy scores from an energy landscape. This energy landscape requires useful features that encode the history of the signal to shape it. A fixed ESN/LSM encodes the history of the time-series data into a state trajectory, reflecting long-range and nonlinear structure. This can then be passed into the EBM, to evaluate whether a possible gap imputation is compatible with the reservoir dynamics. The separation of tasks means that there is no costly backpropagation through time (BPTT) through the reservoir, and training is only done contrastively in the energy readout head. Furthermore, due to the lightweight nature of the readout, the efficiency of inference is improved. Additionally, the EBM provides built-in uncertainty without further calculation required, since the energy rises when a completion seems implausible with reservoir dynamics [1–6]. One important question however lies in the choice of reservoir to produce suitable dynamics for the EBM, which tends to perform better with data that is smooth and Gaussian [7 -9]. In neuroscience and BCI data analysis, neuron population activity is often projected to low-dimensional manifolds, which reveal dominant latent structure and reduce noise [13 -15]. In this study, we apply the same perspective, as although raw ESN/LSM states are high-dimensional, providing more data, applying a PCA to these state trajectories can expose a denoised reservoir manifold that may improve generalisation and interpretability of the time-series by the EBM. We use these two types of reservoirs and these varied activity representations to perform time series imputation at a range of gap lengths, on real ECG recordings centered around the spike to determine the best reservoir representation for this use case. We additionally compare our method against linear interpolation and a second order autoregressive model or AR(2).

## 2 Methods

### 2.1 Datasets and Windowing

We evaluate our models on a singlelead ECG segments drawn from PhysioNet records. We use the Normal Sinus Rhythm Database (NSRDB) records 16265, 16272, 16273, and 16420; resampled to 100 Hz and only the first channel is kept.

We then form fixedlength windows of W = 256 samples with stride S = 128, i.e., 50% overlap. For training the EBM, we compute datasetwide train/validation/test splits in window space with 70/15/15% ratios, respectively

We mask contiguous gaps of length $L \in \{10, 20, 50\}$ samples (i.e., 0.10 s, 0.20 s, 0.50 s at 100 Hz). For ECG, gaps are centered over an Rpeak detected per window. This ensures that our imputation occurs solely over the QRS portion of the ECG trace.

We keep the unmasked values as ground truth and set the masked segment to zero (the downstream pipeline never sees unmasked ground truth).

### 2.2 Baselines

We compare the EBM head to two classical imputers applied only within the gap:

Linear interpolation. For each channel, we linearly interpolate between the gap endpoints if at least two observed samples exist.

AR(2). A secondorder autoregressive model is fit by ridgeregularized least squares (ridge = 1e3) on the linearly prefilled window. Predictions are rolled forward inside each contiguous gap.

### 2.3 Standardized Reservoir Graph

To eliminate topology as a confounder, all reservoir variants (ESNs and LSMs) share a single, standardized adjacency and input map. This contains 256 reservoir units with connection probability = 0.10.

### 2.4 Echo State Networks (ESNs)

We scale the base weight matrix to spectral radius 0.9 and use leaky-integrator dynamics, no bias, and nonlinearities tanh or softsign. Reservoir states (256-dim) are used either raw or PCA-projected, with PCA fitted offline to retain at least 90% variance (min 8 components).

### 2.5 Liquid State Machines (LSMs)

Using the ESN topology, we implement a discrete-time LIF network with 80% excitatory / 20% inhibitory neurons, weights rescaled to 0.7. Parameters: $\tau_m = 20$, threshold = 1.0, refractory = 2, $\tau_{syn} = 5$. Inputs are encoded as 8 level-crossing spike channels. Outputs include binary spikes or smoothed rates (low-pass filtered).

### 2.6 Phase Aware Augmentation

For each window, we augment reservoir features with Hilbert transform features of the input: amplitude and two phase channels ($\cos \phi$, $\sin \phi$). These three channels are standardized per sequence and concatenated to the reservoir states; all models use this augmentation.

### 2.7 Feature Normalization

Before training the EBM, we compute zstatistics over the entire training set:

- For inputs $YYY$ (the raw 1-D signal): $(\mu_Y, \sigma_Y)$.

- For features HHH (reservoir ± Hilbert): $(\mu_H, \sigma_H)$.

We standardize all sequences to zero mean / unit variance with small floors to avoid division by nearzero standard deviations; the same statistics are used for validation/test windows.

## 2.8 Energy Based Head (EBM) and Losses

The EBM is a low-order state-space predictor over reservoir features and the scalar sequence. It maintains a latent state updated with $\kappa = 0.3$, with dynamics parameterized by a stable diagonal plus a low-rank correction (rank 4). Input and emission matrices are dense, and all states/predictions are clamped to $\pm 8$. The per-sequence energy combines prediction error and state consistency, with learned scales and $\gamma_{state} = 1/F$, capped for stability.

We pre-fit the emission and transition matrices with ridge regression and SVD initialization, then stabilize each step by clipping spectral radius, norms, and log-scale parameters. The training objective combines (i) contrastive energy with margin 2.0, (ii) decoder MSE, (iii) short-horizon teacher forcing (1–3 steps), and (iv) small L2 penalties. Loss weights are 0.4 for decoder and 0.4 for teacher forcing, added to the contrastive term. Models are trained with Adam (lr 3e-4, weight decay 1e-5) for 8 epochs, batch size 64, selecting by lowest validation loss.

## 2.9 Imputation Procedure

For our timeseries, we standardize the series, compute forward and backward reservoir features, and roll out the EBM head from both directions. Predictions are blended with a triangular weight schedule peaking at the gap center, with $(\beta_{min}, \beta_{max})$ tuned per dataset. The final fill is the blended output, de-standardized to the original scale.

## 2.10 Architectures Evaluated

We study six reservoirs with different underlying dynamics feeding the same EBM head:

**Table 1.** The types of reservoirs evaluated in this study.

| Variant | Details |
|---|---|
| ESN-tanh-raw | 256 units, $\rho = 0.9$, $\tau_m = 20$, raw states |
| ESN-tanh-pca | Same as above, with PCA to $\geq 90\%$ variance, $\geq 8$ PCs |
| ESN-softsign-raw | Same as ESN-tanh-raw, but with softsign activation |
| ESN-softsign-pca | Same as ESN-softsign-raw, with PCA |
| LSM-spikes | LIF spikes into EBM; level-crossing with $K = 4$ |
| LSM-rates | Low-pass rates into EBM; $\tau_{syn} = 5$ |

Table 1: The types of reservoirs evaluated in this study.

All six share the same connectivity matrix and differ only in neuron model / state postprocessing. All models were trained on an 11th Gen Intel(R) Core(TM) i7-1185G7 at 3.00GHz.

## 2.11 Statistical Analysis

We compute pairwise statistical comparisons between models using standardized mean differences (dz) and Cliff's delta as effect size measures. Raw p-values were obtained through non-parametric tests and subsequently adjusted using Holm's correction to control for multiple comparisons. Statistical significance is indicated with asterisks: $p < 0.05$ (*), $p < 0.01$ (**), $p < 0.001$ (***).

# 3 Results

## 3.1 Reservoir states and PCA manifolds

First we look at the reservoir activity for each of the ESNs and the lower dimensional manifolds that are created form these.

### 3.1.1 LSM

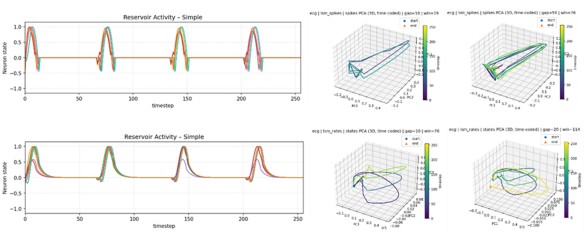

Fgure 1: The above shows the binary spiking of each neuron converted into a trace from the LSM where we fed the raw activity into the EBM. The top right shows this activity after a PCA has been applied. The bottom row of the plot shows the reservoir activity converted into rates using an exponential filter . The bottom right row of the plot shows these rates after a PCA have been applied.

### 3.1.2 ESN

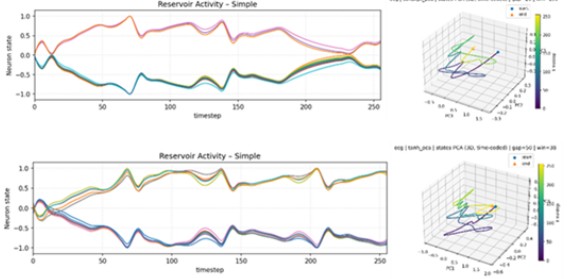

Figure 2: Above shows the ESN reservoir trace and PCA manifold for the ESN with a softsign activation function and below shows the trace and PCA manifold for the ESN with the tanh function.

We can see from Figure 1 that the internal states of the reservoir are able to model the signal waveform to some degree, with the raw activity modelling the QRS complex slightly better, shown by the depression in the wave before the peak, followed by another depression. The PCA manifold shows an interesting recurrent pattern, spanning the first 3 principle components showing a similar pattern of a spike through time. The LSM spike turned rates trace shows a similar pattern, and when the PCA is performed, the manifold is expanded, and follows a more circular recurrent pattern. Figure 2 gives us a different picture, as both ESN type form trajectories that quickly diverge from their start. The pattern of the trajectories shows considerable modulation, but the diversity in the neuronal activity (as with the LSM) is low, with all of the neurons following the either the positive or negative trajectory. The PCA manifolds of these spaces thus show an interesting pattern which does not show much recurrency, with this low dimensional trajectory spanning different parts of the 3D manifold but with no discernible structure. This however only shows the first 3 principal components for graphical representation.

## 3.2 Signal Imputation

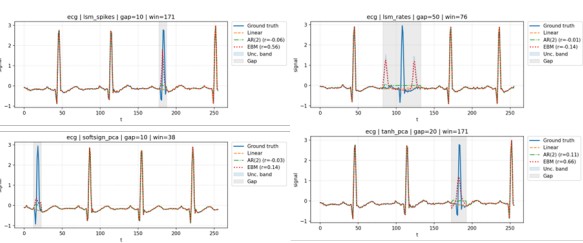

Figure 3: Characteristic examples of ECG reconstruction from the best and worst performing models. On the top left, we have the output from the LSM spike EBM model, the top right is an example from the LSM rates model. The bottom left is the softsign PCA model (worst performing) and the bottom right is the Tanh PCA EBM model.

An interesting observation is that with almost all models and gap sizes successful modelling of spikes always appeared after at least 2 ECG spikes in the trace (successful ECG modelling came on spike 3) while our early and valley spikes consistently appeared on spike number 2, while the first spike was never modelled correctly. This indicates that the EBM potentially had insufficient data to be able to model the incoming data, but even after one spike, it was able to 'know' that it needed to produce a spike and after 2 spike samples it was generally able to produce the spike within the right time frame albeit with lower amplitude. These can be found it he Appendix. This speaks to the recurrent and fading memory nature of the reservoir – more samples into the reservoir creates more consistent patterns that the EBM can use. We hypothesise that with more samples, these reservoir EBMs would be able to more accurately model this. The ESN softsign raw did show this pattern but and reproduced spikes were generally much less pronounced than with the LSM models and ESN tanh.In general, the amplitude of the reconstructed signal was greater when the EBM was using the PCA trajectory over the raw states, which is suitable in this particular usecase. This indicates the PCA removed noise which contributed to better reconstruction.

## 3.3 EBM Model Comparison

| gap | A | B | N | p_raw | dz | cliffs_delta | p_holm | sig |
|---|---|---|---|---|---|---|---|---|
| 10 | lsm_rates | tanh_pca | 235 | 9.03e-38 | -1.14 | -0.246 | 1.36e-36 | *** |
| 10 | lsm_rates | softsign_pca | 235 | 6.14e-36 | -0.865 | -0.185 | 8.6e-35 | *** |
| 10 | lsm_spikes | tanh_pca | 235 | 2.93e-35 | -1.02 | -0.209 | 3.81e-34 | *** |
| 10 | lsm_spikes | softsign_pca | 235 | 1.66e-34 | -0.768 | -0.142 | 1.99e-33 | *** |
| 10 | lsm_rates | tanh_raw | 235 | 1.12e-13 | -0.426 | -0.107 | 1.23e-12 | *** |
| 10 | lsm_rates | lsm_spikes | 235 | 3.34e-13 | -0.236 | -0.0512 | 3.34e-12 | *** |
| 10 | softsign_pca | tanh_pca | 235 | 3.15e-09 | -0.43 | -0.0732 | 2.83e-08 | *** |
| 10 | lsm_rates | softsign_raw | 235 | 6.82e-09 | -0.371 | -0.068 | 5.46e-08 | *** |
| 10 | softsign_raw | tanh_raw | 235 | 8.96e-09 | -0.453 | -0.0467 | 6.27e-08 | *** |
| 10 | lsm_spikes | tanh_raw | 235 | 1.24e-07 | -0.401 | -0.0545 | 7.41e-07 | *** |
| 10 | softsign_raw | tanh_pca | 235 | 1.44e-06 | -0.145 | -0.187 | 7.2e-06 | *** |
| 10 | tanh_pca | tanh_raw | 235 | 0.000189 | 0.0838 | 0.154 | 0.000755 | *** |
| 10 | softsign_pca | softsign_raw | 235 | 0.000275 | 0.0696 | 0.119 | 0.000826 | *** |
| 10 | lsm_spikes | softsign_raw | 235 | 0.0161 | -0.343 | -0.0119 | 0.0322 | * |
| 10 | softsign_pca | tanh_raw | 235 | 0.0381 | -3.7e-05 | 0.0816 | 0.0381 | * |
| … | … | … | … | … | … | … | … | … |

**Table 2.** Pairwise statistical comparisons between reservoir + EBM architectures for ECG signal imputation. Stars indicate significance after Holm correction ($p<0.05$ (*), $p<0.01$ (**), $p<0.001$ (***)).

### 3.3.1 Effect of gap size.

Our analysis revealed that the impact of architectural differences becomes more pronounced as the imputation gap length increases. For short gaps (10 samples), significant differences were observed across models, but the effect sizes were generally modest, reflecting the fact that short gaps are inherently easier to reconstruct. By contrast, at longer gaps (20–50 samples), the effect sizes were substantially larger, demonstrating that the choice of reservoir model has a stronger influence when imputations are more challenging.

### 3.3.2 Model differences.

Liquid State Machine-based reservoirs consistently outperformed Echo State Networks in multiple pairwise comparisons, particularly when imputing longer gaps. Negative dz values in comparisons against ESN-based models indicate that LSM-derived feature streams provided more robust inputs to the energy-based model, enabling more accurate reconstructions. These results underscore the suitability

of spiking-based reservoirs for capturing the nonlinear and sparse dynamics of ECG signals.

### 3.3.3 Raw vs. PCA features.

Applying principal component analysis (PCA) to reservoir states yielded mixed outcomes. In some cases, PCA-projected states improved performance by denoising the high-dimensional representations and enhancing reconstruction accuracy, whereas in other cases raw reservoir states retained richer temporal information that benefited the energy-based model. This suggests that the utility of PCA is task-dependent and may interact with both the reservoir type and the gap length.

### 3.3.4 Conclusion

Overall, the statistical comparisons demonstrate that LSM-based reservoirs, particularly in their raw state representations, offer a consistent advantage in ECG signal imputation, with differences becoming more pronounced as gap size increases. PCA can further refine performance in selected conditions, but its benefits are not universal. These findings highlight the importance of reservoir choice and representation method in designing energy-based imputers for physiological time series.

## 3.4 Uncertainty reliability.

For each dataset $\times$ gap $\times$ architecture, we assesed uncertainty reliability by computing Spearman correlation between mean uncertainty and EBM MAE across windows, and by testing whether mean uncertainty can identify "hard" windows (top MAE quartile) using AUROC. The table below shows these results.

| Gap | Architecture | N | Spearman $\rho$ | AUROC (hard windows) |
|---|---|---|---|---|
| 10 | lsm_rates | 235 | -0.112 | 0.479 |
| 10 | lsm_spikes | 235 | 0.088 | 0.497 |
| 10 | softsign_pca | 235 | -0.275 | 0.471 |
| 10 | softsign_raw | 235 | 0.246 | 0.618 |
| 10 | tanh_pca | 235 | -0.371 | 0.353 |
| 10 | tanh_raw | 235 | 0.363 | 0.672 |
| 20 | lsm_rates | 235 | 0.207 | 0.661 |
| 20 | lsm_spikes | 235 | 0.321 | 0.729 |
| 20 | softsign_pca | 235 | -0.189 | 0.404 |
| 20 | softsign_raw | 235 | 0.333 | 0.762 |
| 20 | tanh_pca | 235 | -0.198 | 0.399 |
| 20 | tanh_raw | 235 | 0.257 | 0.667 |
| 50 | lsm_rates | 235 | 0.414 | 0.802 |
| 50 | lsm_spikes | 235 | 0.624 | 0.828 |
| 50 | softsign_pca | 235 | 0.560 | 0.809 |
| 50 | softsign_raw | 235 | 0.625 | 0.892 |
| 50 | tanh_pca | 235 | 0.341 | 0.644 |
| 50 | tanh_raw | 235 | 0.526 | 0.813 |

**Table 3.** Energy-derived uncertainty reliability (ECG).

## 4 Discussion

From our results we see the EBM's energy-based mean uncertainty being most discriminative when the dynamics are nonlinear or misspecified. However, in smaller gaps the uncertainty can be less informative, as we see AUROC values close to chance level. With respect to feature representation, raw spikes from the LSM-spikes variant were originally assumed to be suboptimal for the EBM head given that they are discrete and sparse. However, both LSM variants consistently produced feature streams that enabled more efficient imputation, particularly for longer gaps where differences between models were most pronounced. PCA-projected states were also found to improve reconstruction in some settings by denoising high-dimensional trajectories, although this benefit was not universal and appeared to depend on both reservoir type and gap length.

We deliberately fixed the ESN, LSM, and EBM parameters to investigate the effect of reservoir type, neuron model, and state representation. However, many studies have shown that parameters such as neuron number, sparsity, and leak rate can have significant implications for reservoir performance. Additional to this, deep reservoirs, which are sequentially ordered reservoirs connected in series, can provide stronger representation power than a single reservoir. Further, the topology of the reservoir has been shown to impact performance, with many studies seeking to determine the optimal topology for the task at hand. These additions were out of scope for the present study but provide ample directions for future work to improve the reservoir–EBM architecture. Future studies could also explore the impact of concatenating multiple reservoir states and applying optimization techniques over reservoir parameters for systematic evaluation.

A further point concerns the parameters we fixed in the EBM head, which directly shape the energy landscape and therefore influence how uncertainty is expressed. We defined energy as quadratic residuals on both prediction error and state consistency, producing elliptical wells aligned with coordinate axes. This stabilises optimization and keeps uncertainty interpretable, but reduces robustness to outliers.

This study demonstrates that combining reservoir computing with an energy-based model provides a robust framework for imputing missing segments in ECG time series while simultaneously quantifying uncertainty. Results show that both LSM- and ESN-based reservoirs produce meaningful latent features, with LSM-based models showing a consistent advantage at longer gap lengths. PCA-projected states often improved reconstruction quality by reducing noise, though their effect was mixed and task-dependent. The EBM's energy scores prove especially effective for longer gaps, offering a reli-

able signal of when imputations may be implausible. Further studies evaluating the minimum amount of data for accurate reconstruction should also be conducted along with optimisation of the reservoir parameters to systematically assess their impact on imputation performance and uncertainty estimation. Overall, this architecture balances efficiency, interpretability, and uncertainty-awareness, making it a promising approach for clinical and physiological data imputation tasks.

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
