# OpenReview forum: "Physiological Signal Imputation using Reservoir Computing Energy-Based Models"
_NLDL.org/2026/Conference — Submitted to NLDL 2026_

### Official Review · Reviewer_iayJ · 2025-09-29
**Good harmony between EBM and Reservoir computing**

**Rating:** 4
**Confidence:** 3

**Summary:**

This work resolve the challenge of imputing missing segments in ECG data, which are often corrupted by sensor dropouts or noise. The authors propose a novel architecture that combines reservoir computing (ESN/LSM) with an Energy-Based Model (EBM). The EBM assigns energy scores to candidate imputations, where low energy indicates good predictions and high energy reflects uncertainty. Reservoir computing expands temporal inputs into rich nonlinear features after dimensionality reduction with PCA. They demonstrated that ESN or PCA-compressed Echo State Network provided the most effective features for accurate and reliable imputation.

**Strengths:**

+ the proposed algorithm well demystified the combination between reservoir computing and EBM.

+ using reservoir computing to figure out complex time series is good choice and this is handed over EBM to impute the ECG signals. Such flow looks natural and works well.

+ Reservoir computing embed low-dimensional time series signal into a rich state sequence making data linearly separable. This is well demonstrated in previous researches and good trials to model complex features underlying time series

**Weaknesses:**

- scalability of dataset : 50,000 records in total.
- diversity of dataset : the proposed algorithm was only demonstrated on ECG dataset.

**Justification:**

This paper addressed that reservoir computings are suitably combined with EBM leading to superior performances on imputations of ECG signals. The authors also provided statistical comparisons on abluation studies(table 2). Finally, the authors concluded that LSM-based reservoirs is best choice for ECG signal imputation with PCA feature extraction. This conclusion was convinced with supported experiments.

---

### Official Review · Reviewer_PtK4 · 2025-09-30
**Nice idea, decent experiments but very terse methods presentation and no evidence of reproducibility**

**Rating:** 2
**Confidence:** 4
**Final Rating:** 2
**Final Confidence:** 4

**Summary:**

This paper investigates the accuracy of reservoir computing architectures, specifically echo state networks (ESNs) and liquid state machines (LSMs), on missing data imputation for echocardiogram data. An innovation of this paper is to combine such architectures with energy-based models which can capture uncertainty. Various combinations of architecture, activation function, and PCA-reductions are investigated. The LSM with leaky-integrate-and-fire neurons appears to be best performing, though results can depend on the level of missingness.

**Strengths:**

Coherence and correctness: As far as it is possible to tell, the implementation appears to be correct. In general the work is well explained and the introduction provides good motivation. Given the space limitations there is a fairly extensive empirical study into the relative performance of different architectures, with a formal attempt to verify that pairs of models give significantly different outputs.

Novelty: The combination of reservoir computing with energy-based models appears to be new. There are clear advantages in this approach in efficient training, avoiding backpropagation, and in providing some measure of uncertainty via energy measurements.
Given the vast design space when it comes to learning problems of this sort, the experiments presented here, while not world-changing, are worthwhile and helpful in providing some guidance on which architectures to use. I did wonder about how far one could make use of the energy-based framework. A main advantage of energy-based models is that one can obtain a probability distribution from an energy E(z) via exp(-E(z))/C. But this probabilistic interpretation does not seem to be used anywhere in this work. Is there any prospect for making statements about your inferences that are more probabilistic?

Clarity and presentation: The paper is well written, especially the introduction which gives a very clear summary and motivation for the work. The standard of writing is generally high. I do have some issues with the presentation of the results (see below).

**Weaknesses:**

Clarity and presentation: Figures are far too small. I zoomed in to 600% and still had difficulty with them. Moreover what is plotted is not fully described in the text: what is a 'reservoir trace'? What do the colors represent? As far as I can tell, shown in Figs 1 and 2 are the values of the network under certain inputs, but is the input for one datapoint or for many? Which ones? Why is the reservoir activity only the same size as the raw data when the motivation for creating a reservoir is to extract higher dimensional features? This is not clearly explained.

A related issue is that the methods description is not given in enough detail. I recognise that there is a strict page limit and authors are encouraged toward brevity, but even so I believe there is room to expand a little (either increasing the length - you still have 1-2 pages free - or with an Appendix. There is mention of an Appendix on line 315 but it does not seem to have been shared with the referees.) For example, I couldn't discern what the reservoir architecture actually is from the two sentences given in Section 2.3. How many layers? What is meant by 'adjacency' in this context? - this terminology does not appear in several of the cited references I checked. What is kappa in Section 2.8? What does 'clamping' mean? What does 'small L2 penalties' mean? There are many many missing details, which might be acceptable if there was either (i) a mathematically detailed appendix, or (ii) freely available code, or (iii) both.

Discussion of limitations: There is some welcome discussion on the mixed results in applying PCA and the relatively poor performance of some architectures. A major omission is any discussion of transformer architectures. I am sure the authors are aware that these have revolutionised many areas of machine learning, and it is conceivable they could be used to address the problem tackled here. Indeed they might alleviate the problem discussed at line 317 regarding the limitations of recurrent memory. This should at least be discussed.

Reproducibility: As noted above, code to reproduce this work is missing. Giving the limited experimental details provided, I think this is quite important.

Minor comments:

- Please provide a reference for PhysioNet and NSRDB.
- Hyphenation throughout the paper seems a little off, with hyphens missing in places: 'secondorder' (162), 'phase aware' (186), 'zstatistics' (194), 'energy based' (202), 'timeseries' (225), 'lower dimensional' (254), 'usecase' (326).
- Typos: 'Fgure' (257), 'but and' (321), 'it he' (314).
- 'when the PCA is performed, the manifold is expanded' (280) - I didn't follow this. PCA is a dimension reduction technique, so what exactly is being expanded?

**Final Justification:**

My main criticisms were a lack of mathematical detail and code limit scope for reproducibility. The authors have offered to rectify this but I am not sure that any neural network architecture can be 'standard' enough to omit all detail and I do not expect to be satisfied by whatever would appear in the final version. I gave a few examples of missing details by asking a series of narrow questions but I didn't mean for these *only* to be taken at face value; I wasn't just requesting answers on these specific things but also hoping to illustrate how much work a typical reader has to do to understand what is going on.

Additionally, the authors did not respond to my question on the usefulness and probabilistic interpretation of using an EBM.

On top of all this another reviewer has raised several valid concerns about a lack of broader context and comparison, which I do not feel were fully addressed by the authors. As it stands, the experiments remain rather parochial. I agree with that reviewer that the methodological contribution is limited and I leave my score unchanged.

**Justification:**

In principle I am not against seeing this work published, but in its current state it is too lacking in enough detail for someone to be able to reproduce the results. Ideally I would like to see both a mathematically complete and rigorous appendix fully describing the experiments, and reproducible code. When both are absent I would be hesitant to accept. The ideas behind the paper otherwise seem sound and it is generally well written.

---

> ### Author Rebuttal · Authors · 2025-10-21
>
> We appreciate the reviewer’s detailed critique and can easily tighten presentation and methods accordingly with larger figures; an expanded Methods paragraph, and are also able to provide our code.
>
> Inevitably, scope choices shaped the comparisons we made. This manuscript is primarily a methodological study, as our aim was to demonstrate how to construct a reservoir-plus-EBM for this specific type of timeseries task under constrained data (we used a very limited subset of the samples) and compute budgets (a laptop CPU), not to pursue a SOTA model. This is also why many of the parameters were fixed - such as the connectivity of the reservoir and EBM structures - to see the impact of changing the reservoir dynamics (activation functions and neuron type). Transformer architectures represent a different modelling paradigm that typically requires substantially larger amounts of data, pretraining, and tuning to be evaluated fairly. Consequently, a direct transformer baseline would change the experiment design and resource assumptions in ways that are outside the paper’s intended aims.
>
> With regarding the meanings of certain words like 'adjacency' matrix - this is the matrix that defines which neuron is connected to which neuron in the reservoir. This is necessary to be fixed as reservoir topology can impact performance.
>
> Regarding the question on 'expanding dimensionality' - there are 100 neurons in the reservoir, each is impacted by the incoming data. This incoming data is 1D whilst the reservoir is 100D - this is what is meant by increasing dimensionality. It is exactly this transformation of data from a 1D to 100D representation that allows linear models to be applied to it and perform well under traditional reservoir computing.
>
> We can additionally add mathematical formulation for ESN, LSM, and EBM update equations, but we actually thought it less pertinent to include here, given they are standard in the literature, but understand it would be useful for those who may not have come across this architecture before.  A table of parameters and their meanings can be added to the appendix, as this is potentially where it is suited best, rather than in the methods
>
> We will correct the typographical and stylistic issues the reviewer flagged (hyphenation, missing references to PhysioNet/NSRDB, and a few typos), and we will reword the sentence that incorrectly stated “the manifold is expanded” during PCA to explain that PCA instead unfolds and denoises the trajectory in the first few principal components.
>
> We will add a brief discussion to make this scope explicit and to note the complementary role of attention- and state-space-based methods for future, as it is the attention mechanism within a transformer that allows it to shine, which can (and has in the literature) been applied to reservoir models, allowing for a lightweight model without a large compute overhead.

---

### Official Review · Reviewer_cWCH · 2025-10-06
**Physiological Signal Imputation using Reservoir Computing Energy-Based Models**

**Rating:** 2
**Confidence:** 3

**Summary:**

This paper proposes a novel framework for imputing missing physiological signals (such as ECG) by combining reservoir computing models (Echo State Networks and Liquid State Machines) as feature extractors with an energy-based model (EBM) as the prediction head. The idea is that the reservoir provides efficient temporal representations without extensive training, while the EBM captures data distributions and generates imputed signals with associated uncertainty. The paper also incorporates PCA-based denoising and forward–backward reconstruction strategies to improve imputation quality. Experimental results on physiological signal datasets demonstrate the model’s potential, especially in terms of computational efficiency and uncertainty estimation.

**Strengths:**

The paper addresses a problem that imputation of missing physiological signals, which is highly relevant in healthcare and wearable device applications. A notable strength is the creative integration of reservoir computing and energy-based models, combining the efficiency of fixed reservoirs with the distributional modeling capabilities of EBMs. This hybrid approach is relatively unexplored and could offer advantages for edge devices where computational resources are limited. The paper is also well-structured, and the proposed pipeline — including PCA denoising and bidirectional reconstruction — shows thoughtful design choices aimed at improving imputation quality and uncertainty estimation. Experimental results support the feasibility of the approach and demonstrate its applicability to real-world physiological data.

**Weaknesses:**

Despite these strengths, the paper has several major weaknesses that limit its novelty and impact. Most of the core components, energy-based models for time-series imputation and reservoir computing for temporal signal processing, are already well-established in the literature. While their combination is relatively new, the paper does not clearly position itself against prior work or provide thorough comparisons with existing EBM-based imputation approaches. The lack of benchmarking against modern state-of-the-art imputation methods (e.g., transformer-based, diffusion-based, or deep generative approaches) also makes it difficult to assess the true value of the proposed method. Furthermore, while the method introduces a new pipeline, it largely assembles known techniques without significant theoretical or algorithmic innovation. The experimental section is also somewhat limited in scope, and ablation studies demonstrating the contribution of each component (reservoir, PCA, forward–backward reconstruction) are missing.

**Justification:**

Although the paper presents a potentially useful combination of reservoir computing and energy-based models for physiological signal imputation, it does not provide sufficient methodological novelty or experimental depth to justify publication in its current form. The individual components of the method are well known, and prior work has already explored energy-based modeling for time-series imputation. The paper does not clearly differentiate its approach from this existing literature, nor does it benchmark against strong contemporary baselines, which raises doubts about the significance of its contribution. Moreover, the experiments, while promising, are limited and lack detailed ablation and comparison studies. Without a stronger demonstration of novelty, more comprehensive evaluation, and clearer positioning relative to prior art, the paper falls short of the standards required for acceptance. Therefore, I recommend rejection in its current form.

However, if the authors strengthen the work by adding comprehensive comparisons with state-of-the-art imputation methods, conducting detailed ablation and component analyses, and clearly articulating how their approach advances beyond existing EBM-based imputation work, the paper could become a valuable contribution and merit acceptance in a future submission.

---

> ### Author Rebuttal · Authors · 2025-10-21
>
> The main aim of this paper was to show how the design choices of the reservoir-EBM impact its performance and representations of the data at different points in the architecture.
>
> We compare 6 reservoir-EBM variants here, divided by how the reservoirs produce their dynamics. The LSMs are spiking neural networks, and the ESNs are recurrent neural networks. We change how the data is represented before feeding it into the EBM with the LSMs, and we change activation functions in the ESNs to create different dynamics before using this in the EBM. As the reservoirs produce high-dimensional data (multiple Poisson-like neural spikes in the LSMs and continuous lines in the ESNs), we further explore this idea of ‘changing the data representation before feeding it into the EBM’. We perform a PCA over the spikes and on the ESN continuous activity to essentially smooth out noise.
>
> This creates several variants of the reservoir-EBM architecture, which we evaluate and show statistical differences in performance. Given the representational power of reservoirs (they can model the shape of data well with sufficient neurons), the internal activity will reflect that. This is what we see in the case of the LSM - the incoming ECG data has been rate encoded into stimulation of the LSM network, and the ECG spike is replicated in neural space, whereas this is not the case in all of the ESNs. This is our reasoning behind showing the LSM and ESN activity plots. Regarding showing the PCA plots - this is another representation that is going into the ESN, it makes sense to visualise this too.
>
> We don’t compare with other EBM models as we are already comparing against 6 variants of our own model and other lightweight models, such as an autoregressive model and linear regression. Reservoir computing is supposed to be lightweight, and our aim was not to develop a model better than other models but to look deeper into this type of model architecture. Ablation would have additionally been useful; however, it was not considered.
>
> A title change may make this clearer, or writing the aim explicitly, I think, would position this better in the eyes of the reader.

---

### Meta-Review · Area_Chair_Xovb · 2025-10-30

**Recommendation:** Reject
**Confidence:** 4

**Metareview:**

The paper proposes a method that combines energy-based models and reservoir computing to reconstruct missing data in biomedical signals.

The reviewers agree that the topic is relevant but found both the empirical evaluation and the methodological contribution to be limited. Moreover, they raised concerns about the quality of the presentation and contextualization in relation to previous work. Overall, these issues make the paper’s contribution to producing new knowledge and advancing understanding insufficiently clear to warrant acceptance.

---

### Decision · Program_Chairs · 2025-11-05

**Decision:**

Reject

**Comment:**

Based on the reviewers and AC comments, the paper cannot be presented at the conference.